# UPLC-MS/MS Analysis of Naturally Derived *Apis mellifera* Products and Their Promising Effects against Cadmium-Induced Adverse Effects in Female Rats

**DOI:** 10.3390/nu15010119

**Published:** 2022-12-27

**Authors:** Alaa Amr, Aida Abd El-Wahed, Hesham R. El-Seedi, Shaden A. M. Khalifa, Maria Augustyniak, Lamia M. El-Samad, Ahmed E. Abdel Karim, Abeer El Wakil

**Affiliations:** 1Department of Zoology, Faculty of Science, Alexandria University, Alexandria 21568, Egypt; 2Department of Bee Research, Plant Protection Research Institute, Agricultural Research Centre, Giza 12627, Egypt; 3International Research Center for Food Nutrition and Safety, Jiangsu University, Zhenjiang 212013, China; 4Pharmacognosy Group, Department of Pharmaceutical Biosciences, Biomedical Centre, Uppsala University, P.O. Box 591, 751 24 Uppsala, Sweden; 5International Joint Research Laboratory of Intelligent Agriculture and Agri-Products Processing, Jiangsu Education Department, Jiangsu University, Nanjing 210024, China; 6Department of Chemistry, Faculty of Science, Menoufia University, Shebin El-Koom 32512, Egypt; 7Department of Molecular Biosciences, The Wenner-Gren Institute, Stockholm University, 106 91 Stockholm, Sweden; 8Institute of Biology, Biotechnology and Environmental Protection, Faculty of Natural Sciences, University of Silesia in Katowice, Bankowa 9, 40-007 Katowice, Poland; 9Department of Biological and Geological Sciences, Faculty of Education, Alexandria University, Alexandria 215260, Egypt

**Keywords:** honey bee, propolis, royal jelly, oxidative stress, reproductive toxicity

## Abstract

Honeybee products arouse interest in society due to their natural origin and range of important biological properties. Propolis (P) and royal jelly (RJ) attract scientists’ attention because they exhibit antioxidant, anti-inflammatory, anti-bacterial, anti-tumor, and immunomodulatory abilities. In this study, we tested whether P and RJ could mitigate the adverse effects of cadmium (Cd) exposure, with particular emphasis on the reproductive function in female rats. In this line, one week of pretreatment was established. Six experimental groups were created, including (i) the control group (without any supplementation), (ii) the Cd group (receiving CdCl_2_ in a dose of 4.5 mg/kg/day), (iii) the P group (50 mg of P/kg/day), (iv) RJ group (200 mg of RJ/kg/day), (v) P + Cd group (rats pretreated with P and then treated with P and Cd simultaneously), (vi) RJ + Cd group (animals pretreated with RJ before receiving CdCl_2_ simultaneously with RJ). Cd treatment of rats adversely affected a number of measured parameters, including body weight, ovarian structure and ultrastructure, oxidative stress parameters, increased ovarian Cd content and prolonged the estrous cycle. Pretreatment and then cotreatment with P or RJ and Cd alleviated the adverse effects of Cd, transferring the clusters in the PCA analysis chart toward the control group. However, clusters for cotreated groups were still distinctly separated from the control and P, or RJ alone treated groups. Most likely, investigated honeybee products can alter Cd absorption in the gut and/or increase its excretion through the kidneys and/or mitigate oxidative stress by various components. Undoubtedly, pretreatment with P or RJ can effectively prepare the organism to overcome harmful insults. Although the chemical composition of RJ and P is relatively well known, focusing on proportion, duration, and scheme of treatment, as well as the effects of particular components, may provide interesting data in the future. In the era of returning to natural products, both P and RJ seem valuable materials for further consideration as anti-infertility agents.

## 1. Introduction

Apitherapy is the science of sustaining health using honeybee hive products. Propolis and royal jelly are essential naturally derived products with a wide range of active biological properties, including antioxidant, anti-inflammatory, anti-bacterial, anti-tumor, immunomodulatory, and anti-infertility effects [1,2,3,4]. Royal Jelly (RJ) is secreted from the hypopharyngeal glands of the worker bees *Apis mellifera*, aged between 5 to 14 days, as a yellowish-white viscous and acidic jelly like substance. Generally, it is a mixture of water, proteins mainly major royal jelly proteins (MRJPs), carbohydrates (glucose, fructose, and sucrose), fatty acids, minerals [potassium (K), calcium (Ca), sodium (Na), zinc (Zn), iron (Fe), copper (Cu), and manganese (Mn)], vitamins (A, B_5_, C, D, and E), and hormones (testosterone, estradiol, and progesterone) [5]. RJ is considered a curative product with high biological and nutritive effects. It alleviates menopausal symptoms by readjusting the hormonal concentration [6], promoting the reproductive performance in polycystic ovarian syndrome in rats [7], ameliorating pregnancy rates [8], counteracting infertility, and reducing oxidative stress [9]. RJ has been shown to improve the integrity of oocytes despite aging [10], and treat asthenozoospermia infertility [11]. Propolis (P) is a resinous product accumulated by honeybees and has a pleasant smell. It consists of resin, wax, essential oils, and organic compounds. Like RJ, it also includes vitamins (B_1_, B_2_, B_6_, C, and E), minerals, and enzymes (glucose-6-phosphatase, succinic dehydrogenase, acid phosphatase, and adenosine triphosphatase) among its components [12]. Studies have shown that P improves semen quality (sperm concentration, sperm motility, and rate of vitality) [13], decreases testicular oxidative stress, and improves the seminiferous tubules structure [14]. It increases fertility by reducing sperm abnormalities and regulates spermatogenesis through rebalancing hormone levels [15].

Reproductive health and fertility are pivotal biological processes affected by lifestyle circumstances, including exposure to environmental toxins. Recently, World Health Organization (WHO) reported infertility as a disease of the reproductive organs that generates disability manifested by a functional impairment [16]. A primary cause of infertility has been associated with defective germ cell production [17,18]. The Centres for Disease Control (CDC) and Prevention reported that male infertility is associated with defects in spermatogenesis, low sperm quality, and anatomical problems [19], while that of female comprises a range of causes affecting ovarian development, maturation of oocytes, tubal obstruction, and impaired implantation [20,21,22]. Furthermore, oxidative stress has been reported as a defective generative function that significantly impacts the reproductive lifespan [23]. Previous studies have correlated the relationship between reproductive toxicity and reactive oxygen species (ROS) exposure. This correlation contributes to sperm DNA damage, induces apoptosis and lipid peroxidation [24,25,26] In the female reproduction system, ROS causes oocyte damage and inhibits ovarian development leading to endometriosis, polycystic ovary syndrome, and subfertility [27].

The relationship between the antioxidant status of serum and seminal plasma and the heavy metal levels in blood has been extensively studied [28,29,30]. Cadmium (Cd) is one of the significant environmental toxicants in the form of cadmium oxide (CdO), cadmium chloride (CdCl_2_), or cadmium sulfide (CdS). According to the substance priority list issued by the Agency for Toxic Substances and Disease Registry (ATSDR), Cd is ranked the 7th most hazardous substance for living organisms [31]. It is a by-product of the processing of other metals such as zinc, lead, or copper, and is used in batteries, glazes, paint pigments, electrochemistry, and agriculture [32]. Cd is ingested in water and food, as in cereals, root vegetables, leafy vegetables, fruits, and seafood [33], and it enters the body through the digestive tract and lungs. Exposure to exogenous chemicals can produce free radicals that exceed the protective effect of the antioxidants, leading to oxidative stress. Cd inhibits the activity of tissue antioxidant enzymes by increasing ROS. It impacts public health and particularly the reproductive system.

This study, therefore, aimed to analyze the chemical profile of P and RJ using linear ion trap-ultra-performance liquid chromatography-tandem mass spectrometry (LTQ-UPLC-MS/MS). The potential effect of these naturally derived *Apis mellifera* products on adverse Cd-induced female reproductive toxicity in rats was investigated. The effect of the administration of RJ (200 mg kg^−1^ BW) or P (50 mg kg^−1^ BW) against a 4.5 mg CdCl_2_ per kg BW over a 30-day exposure period was observed. We evaluated the histopathological and ultrastructure aberrations in the ovarian tissues of Cd-exposed rats and compared the findings to animals receiving P or RJ alone or in combination with Cd. Moreover, oxidative stress markers and the estrus cycle were assessed.

## 2. Materials and Methods

### 2.1. Chemicals

CdCl_2_ was obtained from Sigma-Aldrich (St. Louis, MO, USA). RJ was obtained from Stakich Inc. (Troy, MI, USA), while P was purchased from Apimab laboratory (Avenue du Lac, Clermont l’hérault, France).

### 2.2. Analysis of the Samples Using LTQ-UPLC-MS/MS

Extracts of P and RJ were analyzed using Linear Ion Trap-ultra-performance liquid chromatography-tandem mass spectrometry (LTQ-UPLC-MS/MS) as described by Abosedera et al. [34]. A Shimadzu LC-10 HPLC with a Grace Vydac Everest Narrowbore C18 column (100 mm × 2.1 mm i.d., 5 µm, 300 Å). LC-MS, connected to an LCQ electrospray ion trap MS (Thermo Finnigan, San Jose, CA, USA) was utilized with a mass range of 200–2000 m/z. The raw data files were converted to mzXML format using MSConvert from the ProteoWizard suite (https://proteowizard.sourceforge.io/download.html, accessed on 25 December 2022). The spectra in the network were then searched against Global Natural Products Social Molecular Networking (GNPS) spectra libraries and published data [35,36].

### 2.3. Animals

Thirty 2- to 3-week-old healthy female Wistar albino rats were used in the current study. Animals were obtained from the Faculty of Medicine, Alexandria University, Egypt. They were maintained at the animal house in El Mowasah Educational and Medical Complex, Alexandria, Egypt. They were housed in clean polypropylene cages and maintained under controlled standard laboratory conditions (24 ± 3 °C, 40–60% humidity, 12 h dark/light cycle). They had access to a standard diet and were supplied with tap water ad libitum. Rats were acclimatized to the laboratory environment for two weeks prior to the start of the experiment. The protocol for this study was approved by the n°: 04.21.12.25.2.02 by the Institutional Animal Care and Use Committee (IACUC) at Alexandria University.

### 2.4. Experimental Outline

After the acclimatization period, female (weighing 183.26 ± 13.59 g) rats were randomly divided into six treatment groups of five animals each as follows: (i) control group, 1 mL of distilled water; (ii) Cd group, CdCl_2_ (4.5 mg/kg/day); (iii) P group, 50 mg/kg/day dissolved in distilled water; (iv) RJ group, 200 mg/kg/day dissolved in distilled water; (v) P + Cd group, rats were pretreated with P (50 mg/kg/day) for one week before receiving CdCl_2_ simultaneously with the treatment after the eighth day; vi) RJ + Cd group, animals were pretreated with RJ (200 mg/kg/day) for one week before receiving CdCl_2_ simultaneously with the treatment after the eighth day. Animals received treatment by oral gavage daily for 30 days. The rats’ body weight was monitored at the beginning of treatment and at an interval of 7 days until the end of the experiment.

### 2.5. Sampling and Reproductive Organ Collection

At the end of the experiment, roughly 24 h after the last dose administration, animals were anesthetized using diethyl ether. Before sacrifice, the body weight was recorded. Blood was collected in test tubes by cardiac puncture. Centrifugation was performed at room temperature at 5000 rpm for 5 min to separate serum. The sera were isolated and stored at −20 °C until biochemical analyses were performed. The right ovary was kept at −80 °C for biochemical analysis. A portion of ovarian tissue was homogenized in phosphate-buffered saline (PBS), centrifuged for 15 min at 15,000 rpm, and then the clear supernatants were collected for analysis. Pieces from the left ovary were fixed in 4% formaldehyde and 1% glutaraldehyde in 0.1 M PBS for 3 h for the histological study.

### 2.6. Energy Dispersive X-ray (EDX) Measurement

The content of Cd was determined using EDX microanalysis. Ovaries were frozen at −70 °C and lyophilized at −35 °C for 24 h. The specimens were dried in a carbon dioxide critical point dryer, mounted in aluminum stubs, and coated with a thin layer of gold (≥20 nm) by a JFC-1100E-JEOL Ion sputter evaporator. The specimens were analyzed and photographed in the Electron Microscopic Unit at the Faculty of Science, Alexandria, Egypt, using the Scanning Electron Microscope at an accelerating voltage of 20 kV (JEOL JSM-5300, Tokyo, Japan).

### 2.7. Electron Microscope Examination

For ultrastructure studies, tiny pieces of the left ovary were washed, post-fixed in 1% osmium tetroxide, dehydrated in an ascending grade of ethanol, passed through with propylene oxide, and embedded in Epon resin. Semi-thin sections (1 μm thick) were cut with glass knives using an ultramicrotome (EM UC7, Leica, Wetzlar, Germany) and stained with toluidine blue. Sections were examined by compound microscope (Olympus^®^, Japan) and photographed using a mounted digital camera (Nikon^®^). Ultrathin sections (0.6–0.7 μm thick) were cut with a diamond knife and placed on grids, stained with uranyl acetate and lead citrate. Sections were analyzed and photographed in the Electron Microscopic Unit at the Faculty of Science, Alexandria, Egypt, using the Transmission Electron Microscope (JSM1400-PLUS-JEOL, Tokyo, Japan).

### 2.8. Biochemical Analyses

#### 2.8.1. Thiobarbituric Acid Reactive Substances (TBARS)

The malondialdehyde (MDA) concentration, which is the major end product of lipid peroxide (LPO), was determined by application of the Draper and Hadley method [37] via spectrophotometric measurement of the color produced during the reaction of MDA with TBARS for 30 min at 95 °C in an acidic medium. The intensity of the resulting pink color was correlated to the concentration of MDA and measured using a spectrophotometer at an optical density of 532 nm. The concentration of MDA in each sample was obtained from a standard curve. MDA concentration is reported as nmol/mg protein in tissue samples.

#### 2.8.2. Inflammatory Parameter

Myeloperoxidase (MPO) activity was determined using Myeloperoxidase Colorimetric Activity Assay Kit (Cat no MAK068, Sigma-Aldrich, St. Louis, MO, USA) according to the manufacturer’s instructions. In this method, the MPO enzyme catalyzes the production of hypochlorous acid, rapidly reacting with taurine to form stable taurine chloramine. Finally, taurine chloramine reacts with the yellow thionitrobenzoic acid (TNB) to form a colorless product, dithionitrobenzoic acid (DTNB), and absorbance is measured spectrophotometrically at 412 nm. MPO activity is reported as U/mg protein in tissue samples.

#### 2.8.3. Antioxidant Parameters

Superoxide dismutase (SOD) activity was estimated using the EnzyChrom™ Superoxide Dismutase Assay kit (Cat no EC1.15.1.1, BioAssay Systems, Hayward, CA, USA) according to the manufacturer’s instructions. SOD activity determination relies on the ability of SOD to inhibit the phenazine methosulphate-mediated (PMS) reduction of nitroblue tetrazolium (NBT) dye, and absorbance is measured using a spectrophotometer at 460 nm. SOD enzymatic activity was calculated from the SOD standard curve and is reported as U/mg protein in tissue samples.

Glutathione (GSH) content was measured using Reduced Glutathione Colorimetric Assay Kit (Cat no E-BC-K030M, ELABSCIENCE Company, Houston, TX, USA) according to the manufacturer’s instructions. In this method, a yellow compound whose absorbance is detected spectrophotometrically at 405 nm was produced as a result of DTNB reduction by GSH. The concentration is reported as nmol/mg protein in tissue samples.

Total antioxidant capacity (TAC) activity was determined using the Total Antioxidant Capacity Colorimetric Assay Kit (Cat no ab65329, Abcam, Cambridge, UK) according to the manufacturer’s instructions. TAC concentration was calculated from the Trolox standard curve measured at an optical density of 570 nm. TAC activity is reported as mM/mg protein in tissue samples.

### 2.9. Evaluation of Estrus Cycle

After treatment, the estrous cycle was evaluated. Vaginal swabs were collected daily between 9:00 AM and 10:00 AM for 14 consecutive days. It was obtained as follows, 100 µL of PBS was drawn into the micro tips with the micropipette. The micro tip was gently inserted into the vagina and then transferred to a glass slide. The slide was air-dried, stained with Papanicolaou stain, and examined under a light microscope (CX31, Olympus, Japan). The estrous cycle is divided into 4 phases: proestrus, estrus, metestrus, and diestrus. The vaginal cells are leukocytes, nucleated epithelial cells, and cornified epithelial cells. The proportions of these cells were used to identify each phase. The number of days between the estrus and proestrus phases was used to calculate the estrous cycle length [38].

### 2.10. Statistical Procedures

Statistical analyzes were started by checking the ANOVA assumptions. The Kolmogorov–Smirnov and Lilliefors tests (to determine compliance with the normal distribution) and the Levene test (to control the homogeneity of variances) were performed. These tests allowed for the use of the one-way analysis of variance. The HSD Tukey test was selected as the post hoc test. The results were presented in the charts by means and SD. Homogenous groups were marked with the same letters above the bars. In the case of body mass, the red line marked the initial value before starting the experiment. Asterisks indicate significant differences between the initial and final (bar) values. PCA (Principal Component Analysis) was also performed for oxidative stress parameters, and the graphs show the relationships between variables as well as cases. Statistics 13.1. software was used to carry out the analyzes described above.

## 3. Results

### 3.1. Linear Ion Trap-Ultra-Performance Liquid Chromatography-Mass Spectrometry Analysis

P and RJ were analyzed using linear ion trap-ultra-performance liquid chromatography-tandem mass spectrometry (LTQ-UPLC-MS-MS), and a variety of components were found. As shown in Table 1, the chemical profile of RJ was mainly composed of fatty acids, while flavonoids were the dominant components of P. Besides flavonoids, P contains fatty acid, organophosphorus, and phenolic acid. The carbohydrate, maltulose, phthalate ester, 1,2-benzenedicarboxylic acid, and bis(2-methylpropyl) ester were found in both P and RJ. Afterward, molecular networking within the Global Natural Product Social (GNPS) site [39] helped in connecting the mass spectra of molecules based on the similarity of their fragmentation patterns (Figure 1).

### 3.2. Body Weight of Animals

In the present study, apparent effects on the final body mass of animals after daily exposure to 4.5 mg CdCl_2_/kg, or 50 mg P/kg, or 200 mg RJ/kg, or a combination of CdCl_2_ with either P or RJ for 30 days are well noticed compared to the controls (Figure 2). Interestingly, a significant decrease in the body mass of females pretreated with RJ for one week and then simultaneously treated with RJ and Cd was found.

### 3.3. Assessment of the Impact of RJ and P upon Cd Accumulation in the Testicular and Ovarian Tissues

Energy-dispersive X-ray (EDX) analysis revealed the presence of four basic elements in the ovarian tissues of the control (Figure 3A) and the different treated groups (Figure 3B–F), namely, C, O, Na, and S. The percentage of the elements in the female reproductive tissue is shown in Figure 3. No peak characteristic of Cd was detected in the tissues of rats from the control, P, and RJ groups. While in animals subjected to CdCl_2_ at a dose of 4.5 mg/kg, Cd was detected in the tissues after 30 days. The mean percentage of Cd in the ovarian tissues was 0.20 ± 0.08. Interestingly, pretreatment with P at a dose of 50 mg/kg or with RJ at a dose of 200 mg/kg for 30 days causes a remarkable decrease in the mean percentage of Cd in the ovarian tissues compared to the CdCl_2_ treated groups, as shown in Figure 3G.

### 3.4. Histopathological and Ultrastructure Changes in the Ovarian Tissue

The ovarian appearance was normal in the control rats showing regular germinal epithelium surrounding a dense connective tissue capsule or the tunica albuginea. Normal follicular development with the presence of primordial, primary, and secondary follicles was evident (Figure 4A–C). Treatment with either P or RJ alone does not have any particular characteristics on the normal ovarian appearance. However, CdCl_2_ displayed disrupted ovarian architecture with mild to severe degenerative features, including irregular thickened germinal epithelium, reduction in the layers of granulosa cells in the secondary follicles, loosed granulosa cells, collapsed zona pellucida, cystic and/or distorted follicles, and oocyte dissolution in rats administered Cd (Figure 4D–F). Animals pretreated with either P or RJ for one week and then treated simultaneously with P or RJ with Cd restored normal ovarian tissue appearance with various stages of follicular development. The ovarian epithelium regained its standard thickness and structure, no degenerative changes in ovaries were reported, and the zona pellucida appeared normal (Figure 4G–L).

Ultrastructure features of the ovarian tissue from control rats exhibited healthy follicles. Oocytes contained within the follicles displayed mitochondria with intact cristae and a well-developed rough endoplasmic reticulum. Normal zona pellucida, which is tightly opposed to both the oocyte and the surrounding granulosa cell layer, was observed (Figure 5A–D). Like the semithin sections, the ultrastructure of the ovarian tissues treated with either P or RJ alone manifested no characteristic changes, whereas Cadmium administration mediated aberrations in the ovarian tissue comprising fewer and smaller oocytes, deformed granulosa cells, and follicles with thick theca cell lining. Moreover, compared with the ovarian tissue from the controls, the corrugated basement membrane of the follicle, follicular cyst, vacuolation, and follicular space filled with fluid (Figure 5E–H). Meanwhile, in the P + Cd animals (Figure 5I,J), as well as in the RJ + Cd ovaries (Figure 5K,L), there were no morphological defects in the ultrastructure, even though some mild signs of degeneration were observed.

Furthermore, a two-dimensional PCA explained as much as 96.81% of the total variance (Figure 6F,G). The first principal component (PC 1) was of great importance and explained 94.34% of the variance caused by factors implemented during the experiment. PC 1 was positively correlated with MDA and MPO, whereas it was negatively correlated with TAC, SOD, and GSH. Generally, P or RJ treatment did not significantly influence the measured parameters, creating clusters overlapping the control group (Figure 6G). Rats receiving Cd have significant changes in oxidative stress parameters, manifested as a distinctly separate cluster on the opposite axis side than the control group. Co-treatment with P or RJ and Cd alleviated the adverse effects of Cd, transferring the clusters for combined groups toward the control group. However, still, clusters for co-treated groups (P + Cd or RJ + Cd) were distinctly separated from the control as well as from P or RJ alone treated groups (Figure 6G).

### 3.5. Evaluation of the Effect of P and RJ on Cd-Induced Oxidative Stress and Antioxidant Enzyme Activity in the Ovarian Tissues

Oxidative stress, resulting from the imbalance between reactive oxygen species (ROS) production and antioxidant defense inside the animal, is a risk factor playing a pathogenic role in female infertility.

Regarding oxidative stress biomarkers, exposure of rats to P or RJ alone caused no changes in the total antioxidant capacity (TAC) levels and the malondialdehyde (MDA) in the ovaries supernatants compared to the controls, as shown in Figure 6A,B. However, exposure of rats to Cd as CdCl_2_ was associated with increased production of free radicals leading to a drastic decrease in the level of TAC and a highly significant increase in the level of MDA in the ovaries supernatants. In animals pretreated with P or RJ and Cd simultaneously, the levels of these parameters were rescued to values similar to unexposed controls. Similarly, no effects on the activities of the superoxide dismutase (SOD) and the oxidizing myeloperoxidase (MPO), as well as the concentration of the antioxidant glutathione (GSH) in females receiving P or RJ alone. In contrast, Cd exposure resulted in a significant decrease in the activity of SOD and the concentration of GSH versus a remarkable increase in MPO activity. Combined treatment with P or RJ together with Cd caused the values of these parameters to be restored to the control level (Figure 6C–E).

### 3.6. Assessment of the Estrus Cycle Length in the Different Treated Females

The effects of CdCl_2_, P, and RJ on the estrous cycle are shown in Table 2. The control rats demonstrated a regular cycle with an average length of 4.28 ± 0.55 days and a standard duration of each of the cycle phases. There was a significantly prolonged cycle length in females exposed to Cd as CdCl_2_, P, and P + Cd compared to the control animals. There was no significant difference in cycle length between the controls and the RJ-treated rats. Cd administered to rats caused an irregular pattern with extended cycle duration and changed the frequency of cycle phases. Moreover, significant changes in the number of cycles, the cycle length, and the duration of the proestrus and diestrus phases compared to controls were noticed (Table 2).

## 4. Discussion

The health benefits of bee products, particularly P and RJ, are attributed to their variable bioactive components with diverse beneficial effects on health. Our present findings are consistent with previous studies identifying the presence of fatty acid derivatives, namely 10-hydroxy-2-decenoic acid, 10-acetoxydecanoic acid, (11*S*)-hydroxydodecanoic acid, and 3,11-dihydroxydodecanoic acid in RJ [40,45]; whereas neovestitol, catechin and chlorogenic acid in P [42,44]. Like our results, 1,2-Benzenedicarboxylic acid, bis(2-methylpropyl) ester, and maltulose were formerly identified in both P and RJ [50,51]. Both P and RJ can protect against cadmium-induced ovarian dysfunction [52,53]. Similarly, P shows protective effects against ovarian oxidative stress mediated by Cd owned to the presence of flavonoids, including caffeic acid, phenethyl ester, rutin, and quercetin [51]. P catechin, a phenolic compound with diverse beneficial effects on health, is known for its antioxidant properties and remarkably helpful in preventing and treating ovarian injuries [54]. Catechins possess excellent antioxidant properties and are superior to glutathione, vitamin C and flavonoids [55]. Chlorogenic acid, with its antioxidant, anti-inflammatory, and anti-apoptotic properties, displays protective effects against ovarian injury in mice models [56,57].

The current study investigated how P and RJ could alleviate Cd’s adverse effects on female rats. We have therefore monitored the weight of animals throughout the whole experiment to evaluate the general conditions of rats. The impact of different exposure protocols in the experimental groups had a negative effect on body weight. However, daily food intake did not differ significantly between groups, implying that the reduced body mass observed is unrelated to food intake. Thus, the loss of body mass is most probably the result of increased energy expenditure on defense and/or damage repair. The remarkable weight loss in rats pretreated with RJ for one week and then treated simultaneously with RJ and Cd for 30 days is likely related to the effective role of RJ in initiating the female defense (including immunologic) response against the adverse effects of the nesting Cd. Our findings for body mass are consistent with previous reports [58].

According to the literature, cadmium accumulates in target tissues because its absorption and excretion are slow [59], causing significant oxidative stress in these tissues [60]. It has been shown that Cd concentrations in the ovaries increased significantly [61], demonstrating both direct and indirect effects of CdCl_2_. It is direct because Cd replaces Ca^+2^ and Zn^+2^ in cells by mimicking their physiological processes [62]. It is indirect because Cd in non-reproductive glands, such as the hypothalamus and pituitary, impairs reproductive function by suppressing follicle stimulating and luteinizing hormones release [63]. Herein, pretreatment with P or RJ provided better protection against Cd accumulation. According to our findings, two mechanisms are likely involved in the suppression of Cd accumulation in the ovaries following P or RJ pretreatment. Most likely, investigated honeybee products can alter CdCl_2_ absorption in the gut and/or increase its excretion through the kidneys.

The toxic effects of cadmium on rat ovaries were reflected in histoarchitecture and ultrastructure variations by arresting their development, which was primarily manifested as a decrease in mature follicles and an increase in atresia follicles. Furthermore, these findings demonstrated that cadmium exposure negatively affected the female reproductive system. It also served as a reminder that changes the oocyte microenvironment impacting the quality of developing oocytes and embryos. It was proved that Cd administration leads to degeneration of corpus lutea, oocytes, and granulosa cells. The rarefaction of the granular layer of the follicles observed after Cd administration may lead to disturbances in sex hormones secretion [64]. Our findings suggest that P, like RJ, can significantly improve cellular membrane and organ function, bringing these variables under control. These findings demonstrated that pretreatment of Cd-exposed rats with P reduced Cd’s cytotoxic effects on rat ovaries.

ROS are produced in cells as a result of normal oxidative biochemical reactions, however, the process can be significantly accelerated by external factors. Variations in ROS levels are important in female reproductive functions such as oocyte maturation, ovarian steroidogenesis, follicular development, and ovulation. Importantly, excess ROS production may negatively affect cell function and increase the risk of ovarian pathophysiology [65]. Likewise, our results showed that Cd exposure induced a prominently increased ROS level in the oocytes. They revealed a significant increase in MDA, MPO, and a corresponding decrease in TAC, GSH, and SOD following cadmium exposure. As previously reported, the balance between lipid peroxidation and antioxidants indicates oxidative stress status. The ovarian TAC level was significantly lower in the Cd animals compared to the controls, which is consistent with other studies [66]. This low level of TAC might explain the high ovarian MDA level in the Cd-treated rats. However, the ovarian TAC level in the P + Cd and RJ + Cd groups was significantly higher than in the Cd group, which might be attributable to the low ovarian MDA level found in the P + Cd and RJ + Cd groups. Pretreatment with P or RJ significantly prevented oxidative stress in the ovaries of rats, as seen by the increased SOD activity. Additionally, GSH concentration decreased significantly in the ovaries of rats treated with CdCl_2_. GSH is a non-enzymatic antioxidant known to scavenge free radicals by direct interaction of its thiol group with ROS, and it acts as a substrate for both GST and GPx.

The induction of cadmium is known to affect the hypothalamus-pituitary-ovarian axis and decrease follicle-stimulating hormone, luteinizing hormone, and progesterone levels, which are responsible for ovarian development and ovulation. Our study revealed a significantly low percentage of the regular estrus cycle, which agrees with previous studies [64]. Furthermore, this finding suggests that RJ administration has the potential to modulate estrogenic activity, which could help to improve the impaired reproductive function seen in polycystic ovarian syndrome.

## 5. Conclusions

Our research demonstrates that pretreatment with P or RJ can perfectly prepare the organism to overcome harmful factors. Undoubtedly, all the beneficial effects described above encourage further research, which can enable a thorough understanding of the mechanism of action of these honeybee products. These effects are undoubtedly due to the rich composition of RJ and P. Although their chemical composition is already known, focusing on proportion, duration, and scheme of treatment, as well as the effects of particular components, may provide interesting data in the future. In the era of returning to natural products, both P and RJ seem valuable materials for further exploration.

## Figures and Tables

**Figure 1 nutrients-15-00119-f001:**
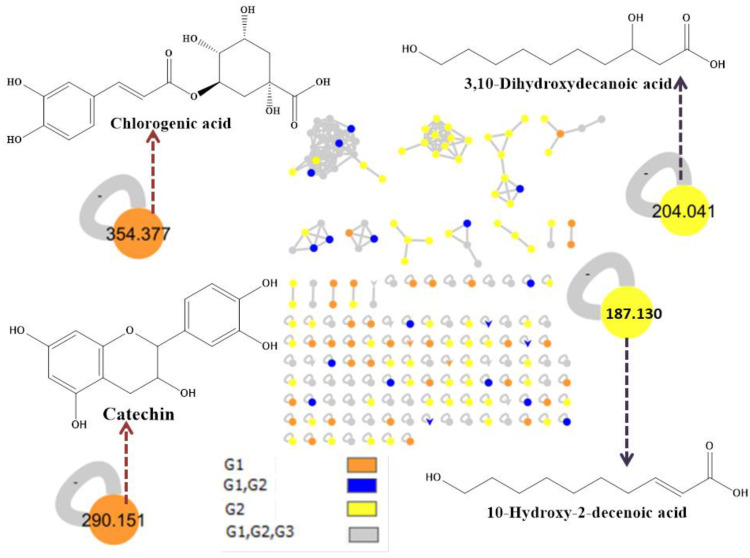
Identified bioactive compounds from P and RJ using LQT-MS-MS analysis and assisted Global Natural Product Social (GNPS) molecular network. The nodes refer to parent masses of the extracts metabolites. The circular nodes refer to the whole parent masses. The triangle nodes represent parent ions that have been identified in the GNPS molecular network. G1: Propolis, G2: Royal jelly, G3: blank solvent used.

**Figure 2 nutrients-15-00119-f002:**
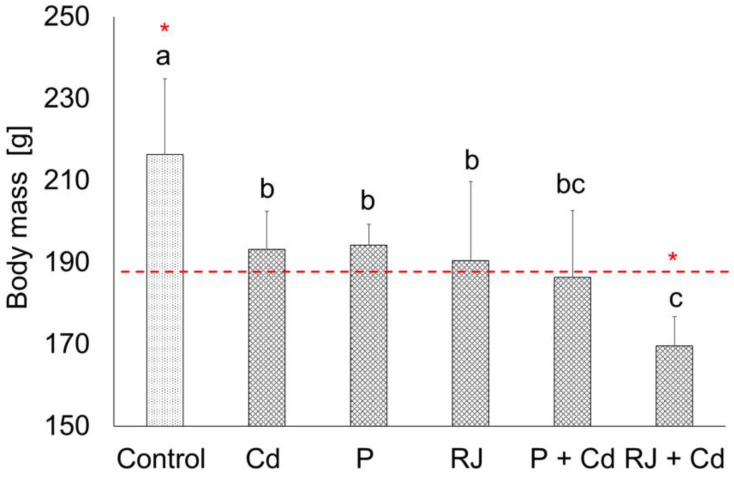
Final body mass [g] of females after exposure to Cd, P, RJ, and a combination of P or RJ with Cd (P + Cd or RJ + P). The same letters denote homogenous groups. The asterisks above bars indicate significant differences between the initial (red line) and final (bar) body mass in each group (ANOVA, HSD Tukey test, *p* < 0.05). The average initial body mass was 183.27 ± 13.36. Cd—cadmium group where rats were receiving Cd as CdCl_2_ (4.5 mg/kg/day) during the experiment, P—propolis group where animals were treated with propolis (50 mg/kg/day), RJ—royal jelly group where animals were treated with royal jelly (200 mg/kg/day), P + Cd—combined treatment group where animals were pretreated with propolis for one week and then treated simultaneously with propolis and cadmium, RJ + Cd—combined treatment group where animals were pretreated with royal jelly for one week and then simultaneously treated with royal jelly and cadmium.

**Figure 3 nutrients-15-00119-f003:**
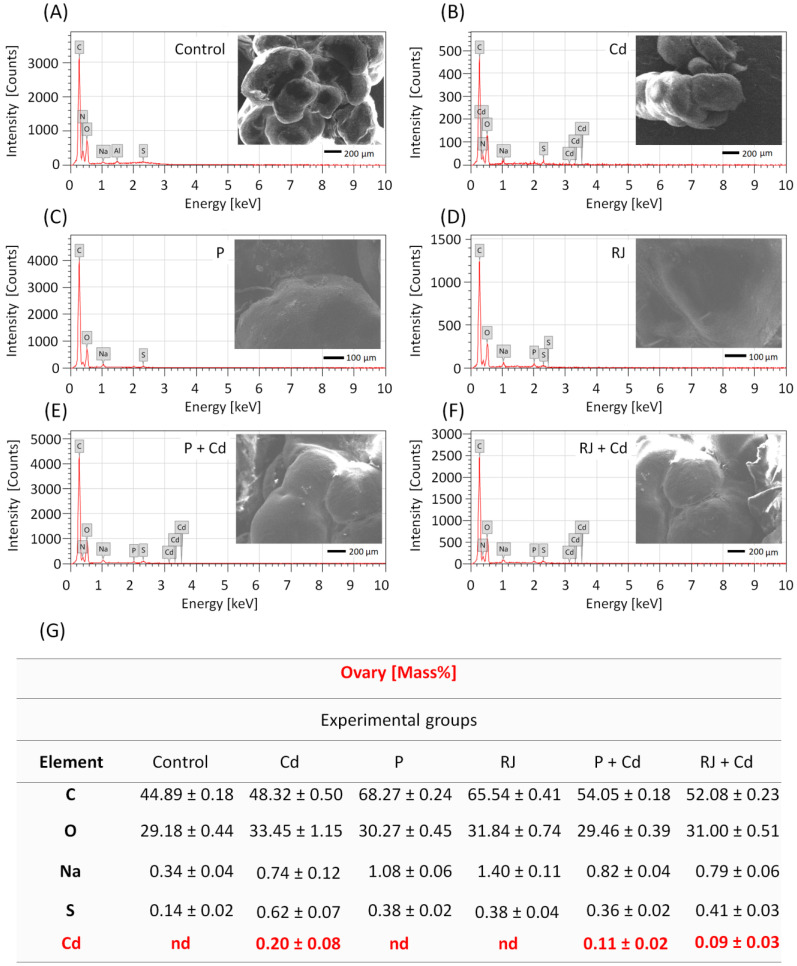
EDX spectra of a selected area of the ovary (SEM of the tissue inserted in the graphs) in the different treated groups: control (**A**), Cd (**B**), P (**C**), RJ (**D**), P + Cd (**E**), and RJ + Cd (**F**), illustrating qualitative elemental composition, and quantitative analysis (**G**). In Figures (**A**–**E**): horizontal scale—X-ray energy; vertical scale—X-ray counts. C—carbon, Na—sodium, O—oxygen, S—sulfur, Cd—cadmium, nd—not detected.

**Figure 4 nutrients-15-00119-f004:**
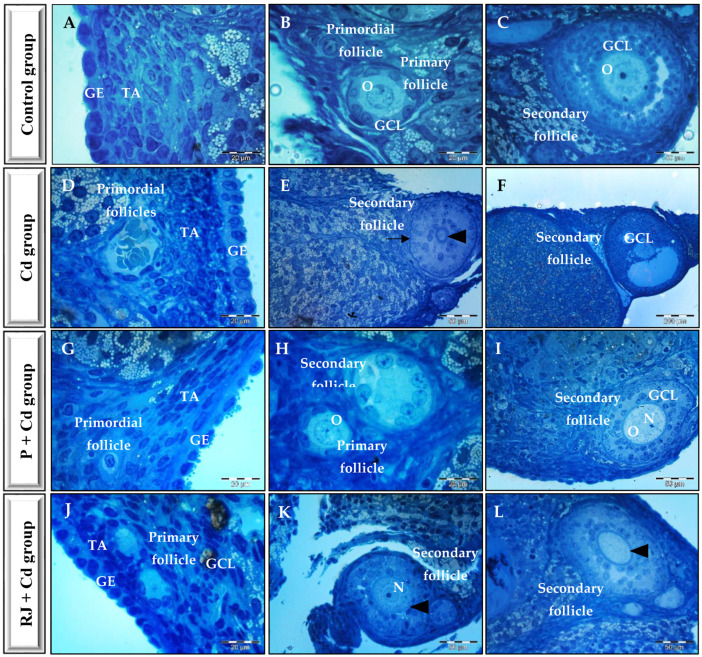
Photomicrographs of toluidine blue stained sections of ovaries from different experimental groups. Panels (**A**–**C**) show normal histoarchitecture of ovary; Panels (**D**–**F**) illustrate Cd-induced histopathology of the ovary with corrugated basement membrane (arrow), a relatively thick zona pellucida (arrowhead), fluid–filled space (asterix), and a degenerated granulosa cell layer at (**F**); Panels (**G**–**I**) show the architecture of ovary from animals pretreated with P for one week and then treated simultaneously with P and Cd; Panels (**J**–**L**) show the architecture of ovary from animals pretreated with RJ for one week and then treated simultaneously with RJ and Cd and restore normal zona pellucida appearance (arrowhead). GCL—granulosa cell layer, GE—germinal epithelium, N—nucleus, O—oocyte, TA—tunica albuginea.

**Figure 5 nutrients-15-00119-f005:**
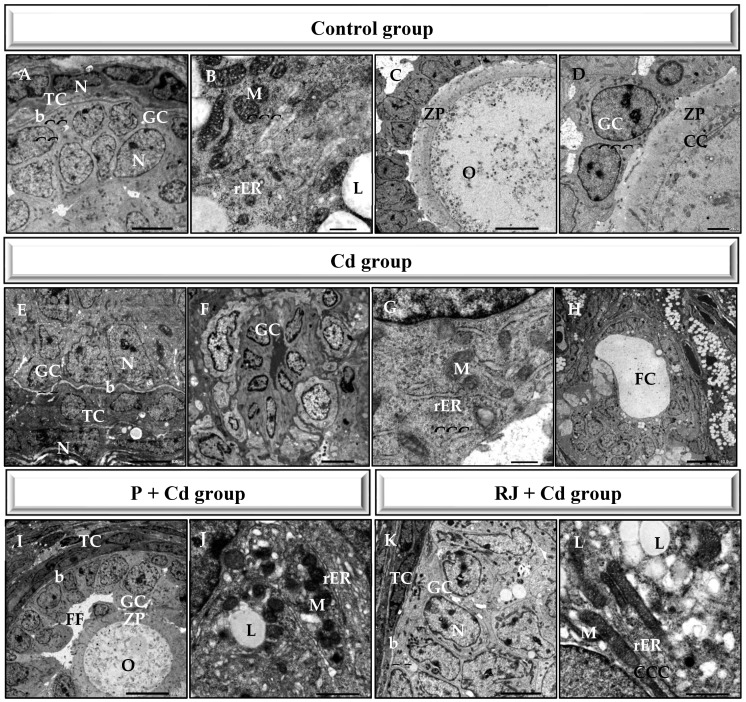
Transmission electron micrographs of ovaries from different experimental groups. b—basement membrane, FC—follicular cyst, FF—follicular fluid, GC—granulosa cell, L—lipid, M—mitochondria, N—nucleus, O—oocyte, rER—rough Endoplasmic Reticulum, TC—theca cell, ZP—zona pellucida.

**Figure 6 nutrients-15-00119-f006:**
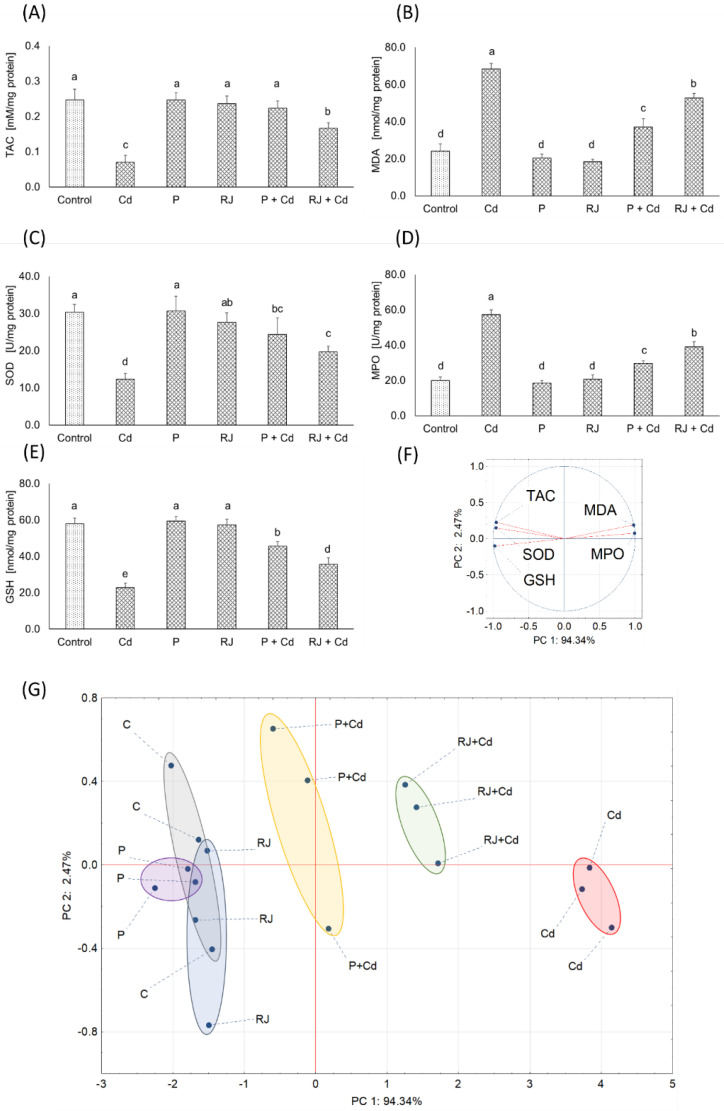
Oxidative stress markers in females’ ovaries. (**A**) total antioxidant capacity (TAC); (**B**) malondialdehyde concentration (MDA); (**C**) superoxide dismutase (SOD) and (**D**) myeloperoxidase (MPO) activity; (**E**) reduced glutathione (GSH) concentration. (**F**) Principal component analysis (PCA) on oxidative stress markers variables, and (**G**) 2D PCA plot of control and experimental group cases. The same letters in each chart, a–e denotes homogenous groups (ANOVA; HSD Tukey test; *p* < 0.05). Shape colors in (**G**) denote: gray—Control group, violet—Propolis group, blue—Royal Jelly group, yellow—Cadmium + Propolis group, green—Cadmium + Royal Jelly group, red—Cadmium group.

**Table 1 nutrients-15-00119-t001:** Bioactive compounds identified from Prolis and Royal jelly using LTQ-UPLC-MS-MS analysis.

No.	Compound	R_t_	MW	MF	MS^2^	Class of the Compounds	Reference
1	10-Acetoxydecanoic acid ^1^	1.17	229.759	C_12_H_22_O_4_	199.9730, 185.0170, 98.9600, 86.0030, 69.999	Fatty acid	[40]
2	3,11-Dihydroxydodecanoic acid ^1^	1.32	232.078	C_12_H_24_O_4_	141.9810, 99.9910, 69.9010	Fatty acid	[40]
3	Catechin ^2^	2.42	290.08	C_15_H_14_O_6_	272.0930, 254.9760, 237.0960	Flavonoid	[41]
4	Neovestitol ^2^	3.10	272.140	C_16_H_16_O_4_	254.2770, 237.0920	Flavonoid	[42,43]
5	Isocupressic acid ^2^	3.80	319.115	C_20_H_32_O_3_	272.0480, 177.0170,	Fatty acid	[44]
6	10-Hydroxy-2-decenoic acid ^1^	4.25	187.1300	C_10_H1_8_O_3_	169.0010, 152.8800, 141.9000, 138.0102, 125.3000, 112.0030, 87.000	Fatty acid	[45]
7	Quercetin tetramethyl ether ^2^	5.91	358.290	C_19_H_18_O_7_	313.1370, 151.2220	Flavonoid	[46]
8	(*2R,3R*)-Pinobanksin 3-isobutyrate ^2^	6.41	328.210	C_19_H_20_O_5_	251.0690, 175.0410, 157.0110,	Flavonoid	[47]
9	Maltulose ^1,2^	6.50	360.099	C_12_H_24_O_12_	324.9800, 276.9570, 258.9660, 162.9560	Carbohydrate	[48]
10	Tributyl phosphate ^2^	6.87	267.561	C_12_H_27_O_4_P	249.0670, 221.9830, 210.9860, 154.8940, 136.0840, 98.8430	organophosphorus	https://bit.ly/3yDP42j (accessed on 12 November 2022)
11	(11*S*)-Hydroxydodecanoic acid ^1^	10.02	216.093	C_12_H_24_O_3_	125.9520, 99.9920, 83.9820, 69.8470	Fatty acid	[40]
12	Chlorogenic acid ^2^	10.50	354.377	C_16_H_18_O_9_	175.0050, 163.1070	Phenolic acid	[49]
13	3,10-Dihydroxydecanoic acid ^1^	13.99	204.041	C_10_H_20_O_4_	169.0100, 83.9380	Fatty acid	[48]
14	1,2-Benzenedicarboxylic acid, bis(2-methylpropyl) ester ^1,2^	14.88	279.010	C_16_H_22_O_4_	232.9580, 223.0710, 219.0410, 204.9220, 201.0100, 172.9940, 166.9990, 148.9170	Phthalate ester	https://bit.ly/3eFS8UC (accessed on 12 November 2022)

^1^: Royal jelly, ^2^: Propolis, MW: molecular weight, MF: molecular formula, MS^2^: Mass spectrometry.

**Table 2 nutrients-15-00119-t002:** Phases of the estrus cycle and its length in the different treated rats.

Group	No. of Cycles	Cycle Length	Days in Each Phase			
			Proestrus	Estrus	Metestrus	Diestrus
Control	3.33 ± 0.47	4.28 ± 0.55	1.25 ± 0.43	1.5 ± 0.50	0.75 ± 0.25	2.67 ± 0.47
P	3 ± 0.81 ^b^	5.05 ± 0.34 *	1 ^a,b^	1.33 ± 0.47 ^b^	0.83 ± 0.24	2.17 ± 0.27 *^,a,b^
RJ	3.67 ± 0.47	4.5 ^a,d^	1 ^a^	1.5 ± 0.41	0.67 ± 0.24	2.17 ± 0.24 *^,a^
Cd	2.67 ± 0.47 *	5.22 ± 0.39 *	1.67 ± 0.47 *^,c^	1.33 ± 0.47 ^b^	0.80 ± 0.24	1.67 ± 0.47 *^,c^
P + Cd	2.33 ± 0.47 *^,c^	5.11 ± 0.63 *	1.5 ± 0.41 ^c^	1.83 ± 0.62 ^a,c^	0.83 ± 0.24	1.83 ± 0.24 *^,c^
RJ + Cd	3.33 ± 0.94	4.16 ± 0.24 ^a,e^	1.16 ± 0.24 ^a^	1.33 ± 0.58	0.67 ± 0.24	2.33 ± 0.47 ^a^

* Statistical significance compared to the corresponding control group, ^a^ Statistical significance compared to the corresponding Cd group, ^b^ Statistical significance compared to the corresponding P + Cd group, ^c^ Statistical significance compared to the corresponding P group, ^d^ Statistical significance compared to the corresponding RJ + Cd group, ^e^ Statistical significance compared to the corresponding RJ group. All values were expressed as mean ± SD, *p* ≤ 0.05.

## Data Availability

Not applicable.

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
