# Peer review of "UPLC-MS/MS Analysis of Naturally Derived Apis mellifera Products and Their Promising Effects against Cadmium-Induced Adverse Effects in Female Rats"

_nutrients, 2022, doi:10.3390/nu15010119_

Round 1

Reviewer 1 Report

Attach file

Author Response

The authors described the results of Propolis and royal jelly effects of cadmium (Cd)
over reproductive function in female rats.
The results obtained are promising.
The results are well discussed and compared with literature data
The compounds identified are from the class of flavonoids and fatty acids.
The Figures and tables are in good quality and with adequate information for dates
presentation.
The statistical significance analysis was careful showing the differences between the
treatments.
The conclusions are coherent with of results obtained.

The manuscript is suitable for publication in Nutrients.

Thank you for the reviewer’s comments upon our manuscript that encourage us.

Minor corrections:
- p.6, line 8: … (2R,3R)-Pinobanksin 3-isobutyrate ... change by .... (2R,3R)-Pinobanksin 3-
isobutyrate ...

Response: Corrected

- p.7, line 1 ... (11S)-Hydroxydodecanoic acid ...change by... (11S)-Hydroxydodecanoic acid ...
p. 14-17:

Response: Changed

check all references

Response: Done

Reviewer 2 Report

1 - The abstract is very well written.

2 - The Keywords were well chosen and represent the work complementing the title in a brilliant way.

3 - In my personal view, I would not abbreviate the words Royal Jelly (RJ) and Propolis (P) because, in my understanding, they are small and important words. This type of abbreviation can be confusing for the reader.

4 - Some studies have shown toxic compounds in propolis. Did this study evaluate this possibility?

5 - In all figures and tables it is necessary to write in the caption all the acronyms, including those that have already been described in the text as P, PJ, Cd (...). Figures and tables need to be self-explanatory.

6 - In general, the results were well explored and presented.

7 - I believe that nutritional data of the samples must be presented (sugars, lipids, proteins...). Couldn't excess consumption bring harm? Did the authors assess the levels of blood glucose and cholesterol in these animals?

8 - The manuscript was, in general, well discussed and the conclusion responds to the objective of the study.

9 - References are current. Attention to reference number 20. I believe there was a typing error.

I congratulate the authors for the excellent work.

Author Response

Dear Sir,

Thank you for the interesting comments dated December 18th 2022, upon our manuscript entitled “UPLC-MS/MS analysis of naturally-derived Apis mellifera products and their promising effects against cadmium-induced adverse effects in female rats.” by Amr et al. (Ref: Submission ID 2116564). We have now addressed the reviewers concerns thoroughly, please see detailed response below. We hereby send you a revised version of our manuscript, in which we have highlighted in red colour additional and/or corrected text and information in answer to the reviewers’ comments.

Answers to the reviewer’s comments.

The responses to the questions of Reviewer # 1 were in red colour.

Reviewer 3 Report

The topic of this article is very interesting. The health effects of bee products have been studied and known for a long time, and investigating them for other health potentials is really important and a possible alternative to drugs.

I think the article is well written and organized.

Minor comments:

Page 2 lines 49, 52-54, 58-59, 63-64, 64-65, 72-73, 82-83, 88-90, 96-99, bibliographic references are missing. Please add them.

Page 2 line 55 in royal jelly there are other proteins (including enzymes), although MRJPs are the main ones they are not the only ones. Please implement the text.

Page 3 line 103 I think the authors meant to write "was investigated".

Page 3 line 131 “ad libutum” should be written in italics. Please correct it.

Page 3 Experimental outline: On what basis were the doses of propolis and royal jelly to be administered to the animals chosen? Why were they pretreated for a week before cadmium was administered? How was the propolis provided? Was it dissolved in a solvent? Please clarify in the text.

Page 5 lines 204, 207 I believe that if capacity is measured, activity is not measured. I think “capacity” is right in this case so I would delete the term “activity”.

Page 5 Statistical procedures: since multiple comparisons are made, I wonder if a correction should be applied (such as Bonferroni correction or others).

Page 8 figure 3 the graphs cannot be read, the numbers and axis titles are too small. Please arrange the graphs differently so that they are enlarged.

Pages 9 and 10 figures 4 and 5 P-only and RJ-only groups are missing. Is there a reason or have they been left out? Please add them.

Author Response

Dear Sir,

Thank you for the interesting comments dated December 19th 2022, upon our manuscript entitled “UPLC-MS/MS analysis of naturally-derived Apis mellifera products and their promising effects against cadmium-induced adverse effects in female rats.” by Amr et al. (Ref: Submission ID 2116564). We have now addressed the reviewers concerns thoroughly, please see detailed response below. We hereby send you a revised version of our manuscript, in which we have highlighted in green colour additional and/or corrected text and information in answer to the reviewers’ comments.

Answers to the reviewer’s comments.

The responses to the questions of Reviewer # 1 were in green colour
